# Analysis of Thioredoxins and Glutaredoxins in Soybean: Evidence of Translational Regulation under Water Restriction

**DOI:** 10.3390/antiox11081622

**Published:** 2022-08-21

**Authors:** María Martha Sainz, Carla Valeria Filippi, Guillermo Eastman, José Sotelo-Silveira, Omar Borsani, Mariana Sotelo-Silveira

**Affiliations:** 1Laboratorio de Bioquímica, Departamento de Biología Vegetal, Facultad de Agronomía, Universidad de la República, Avenida Garzón 780, Montevideo 12900, Uruguay; 2Departamento de Genómica, Instituto de Investigaciones Biológicas Clemente Estable, MEC, Av. Italia 3318, Montevideo 11600, Uruguay; 3Department of Biology, University of Virginia, 485 McCormick Rd., Charlottesville, VA 22904, USA; 4Departamento de Biología Celular y Molecular, Facultad de Ciencias, Universidad de la República, Iguá 4225, Montevideo 11400, Uruguay

**Keywords:** *Glycine max*, thioredoxin, glutaredoxin, drought, transcriptome, translatome, root

## Abstract

Soybean (*Glycine max* (L.) Merr.) establishes symbiosis with rhizobacteria, developing the symbiotic nodule, where the biological nitrogen fixation (BNF) occurs. The redox control is key for guaranteeing the establishment and correct function of the BNF process. Plants have many antioxidative systems involved in ROS homeostasis and signaling, among them a network of thio- and glutaredoxins. Our group is particularly interested in studying the differential response of nodulated soybean plants to water-deficit stress. To shed light on this phenomenon, we set up an RNA-seq experiment (for total and polysome-associated mRNAs) with soybean roots comprising combined treatments including the hydric and the nodulation condition. Moreover, we performed the initial identification and description of the complete repertoire of thioredoxins (Trx) and glutaredoxins (Grx) in soybean. We found that water deficit altered the expression of a greater number of differentially expressed genes (DEGs) than the condition of plant nodulation. Among them, we identified 12 thioredoxin (Trx) and 12 glutaredoxin (Grx) DEGs, which represented a significant fraction of the detected GmTrx and GmGrx in our RNA-seq data. Moreover, we identified an enriched network in which a GmTrx and a GmGrx interacted with each other and associated through several types of interactions with nitrogen metabolism enzymes.

## 1. Introduction

Roots are the first plant organ that experiences drought stress, as they are in direct contact with the soil and sense water limitation [1]. Plants evolved different mechanisms to adapt to drought stress, such as developing wider root systems for water uptake or restricting water loss in the above-ground tissues. Similar to other stresses, water deficit triggers the induction of functional and regulatory genes and secondary messengers such as Ca^2+^ and reactive oxygen species (ROS). An increase in ROS causes oxidative damage to proteins, DNA, and lipids [2] and ultimately triggers a programmed pathway for cell death. However, the standpoint of ROS as a toxic byproduct that requires a scavenging mechanism to protect the cell has been modified in a few years to one that pinpoints it as a key signal transducer in plant biology [1]. Maintaining a basal level of ROS in cells is essential for cellular proliferation and differentiation processes [3]. Plants have many antioxidative systems involved in ROS homeostasis and signaling; these include NOX-like proteins (RBOHs), superoxide dismutases (SODs), catalases (CATs), peroxiredoxins (PRXs), glutathione peroxidase (GPXs), and a network of thio- and glutaredoxins [3].

The thioredoxin superfamily consists of three main subclasses of oxidoreductases: thioredoxins (Trx), glutaredoxins (Grx), and protein disulfide isomerases (PDI) [4]. Most Trx and Grx have the CxxC motif as a common redox center localized on the surface of the redoxin in the three-dimensional folding. Furthermore, *trx* and *grx* mutants show partial and complex redundancies among the Trx and Grx members. Moreover, genetic evidence supported the existence of crosstalk between Trx and Grx systems, which makes plant redox systems of high plasticity [5].

Thioredoxins (Trx) are a multigenic family of small proteins in plants that contain two redox-active Cys residues and maintain redox balance homeostasis through thiol-disulfide exchange reactions [6,7]. Trx uses as electron donor Trx reductases using NADPH or reduced ferredoxin (Fdx) for a new catalytic cycle [8].

Trx of vascular plants are divided into two categories according to the active site, the typical (WCGPC active site) and the atypical (XCXXC active site), which are further subdivided into several classes according to their structure and cellular compartmentation [9]. Typical Trx include seven types: Trx f, h, m, o, x, y, and z [10], depending on their localization within the cell, sequence similarity, and molecular functions. Trx from the f, m, x, y, and z-type have plastidial localization and regulate metabolic process and redox balance in chloroplasts. Trx from the o-type localize in mitochondria, except in pea (*Pisum sativum*), in which they are found in nuclei and mitochondria. The o-type Trx have diverse functions, including regulation of cell cycle progression, stress responses, and activation of the Tricarboxylic-acid cycle (TCA) in the mitochondria [10]. The largest family of Trx is the h-type, with eight isoforms in *Arabidopsis thaliana* (Arabidopsis). Many of them localize to the cytosol, while some to the ER-Golgi membranes and plasma membrane. They present diverse functions depending on the plant species (i.e., in barley and wheat Trx h is involved in seed development, while in Brassica Trx h has a role in self-incompatibility response and as a molecular chaperone to help Arabidopsis plants to cope with heat stress) [4,5,10].

On the other hand, atypical Trx proteins in Arabidopsis are subdivided into nine classes: the cytosolic Trx-h-like protein class, the CDSP32 (chloroplastic drought-induced stress protein of 32 kDa) class, and the ACHT (atypical Cys His-rich Trx) protein class or Lulium Trx. The chloroplastic HCF164 (High Chlorophyll Fluorescence 164) and the chloroplastic TRX-like3 form two independent classes. Furthermore, the nucleoredoxin (NRX) and the cytosolic Clot classes cluster separately [9]. The function of members of these categories awaits elucidation [5,7,9].

In *Medicago truncatula*, there is an additional Trx type, called Trx s, that comprise four isoforms that are associated with symbiosis [11,12].

Likewise, glutaredoxins (Grx) are known to catalyze the reversible reduction of disulfide bonds utilizing the reducing power of glutathione (GSH) and NADPH-dependent glutathione reductase (GR), and have a role in maintaining and regulating cellular redox homeostasis [1].

The phylogenetic analysis based on the similarities and characteristics of the redox center defined five subgroups of Grx: subgroup I—C[P/G/S] Y [C/S]; subgroup II—CGFS; subgroup III—CCx[C/S/G] or ROXY; subgroup IV—4CxxC; and subgroup V—CPF[C/S] [5,13,14,15], which includes the typical Grx (of about 10–12 kDa) as well as other proteins characterized by having multiple Grx domains or a combination of Grx domains associated with other domains. In contrast to the Trx phylogenetic organization, not all the members in a group of Grx have the same subcellular localization. Moreover, not all of the five subgroups of Grx are found in all species of the green lineage: subgroups III and IV are specific to vascular plants [5,15].

Leguminous plants, such as soybean (*Glycine max* (L.) Merr.), can establish symbiosis with rhizobacteria. This mutualist relationship elicits the formation of a new organ within the root, the symbiotic nodule, where the reduction of atmospheric di-nitrogen into ammonia-biological nitrogen fixation (BNF) takes place. The establishment and correct function of the BNF process involves a redox control key for plant–rhizobia crosstalk and nodule metabolism [16]. There is growing evidence of the involvement of the thioredoxin and glutaredoxin systems in the redox control, regulating the redox state of proteins in both symbiotic partners [16]; nevertheless, there is no exhaustive classification of these systems in soybean to date. Moreover, the BNF process is highly susceptible to water-deficit stress [17]. Our group has evidence suggesting that soybean genotypes respond differentially when subjected to water-deficit stress depending on whether or not the plant is nodulated. Furthermore, previous studies have shown that exposing plants to certain environmental conditions causes a global inhibition of translation initiation that is visualized as a decrease in the percentage of polysome-associated mRNA [18]. Moreover, direct analysis of the subset of mRNAs that are being translated allows a more accurate and complete measurement of the cell gene expression than the one obtained when only mRNA levels are analyzed [19,20,21].

To shed light on this phenomenon, we set up a RNA-seq experiment including total RNA (TOTAL) and polysome-associated mRNA (PAR) fractions to analyze the transcriptome and translatome of soybean plants subjected to four combined treatments, including the nodulation and the water-deficit condition.

The aims of this work were: first, to contribute to the soybean Trx and Grx identification and classification; and second, to analyze the transcriptional and translational regulation of differentially expressed Trx and Grx between two combined treatment comparisons, including nodulated and non-nodulated plants and water-restricted and well-watered plants.

A total of 125 Trx and 89 Grx homologs were identified from the *Glycine max* v4.0 proteome. Even though *Trx* and *Grx* show an uneven physical location distribution, there is at least one member of each family on each chromosome. Between them, 12 *Trx* and 12 *Grx* emerged in the differentially expressed gene (DEG) lists between the two combined treatment comparisons analyzed herein. Six of the Trx belong to the most abundant Trx typical class (the h-type), whereas the other six proteins belong to the atypical Trx, comprising nucleoredoxin, TTL, and Lilium classes. The 12 Grx found belong to the 4CxxC and CCxS classes.

## 2. Materials and Methods

### 2.1. Sequence Retrieval and Identification and Initial Characterization of Trx and Grx Family Members

The *Glycine max* v4.0 reference proteome sequences and annotation files (GCF_000004515.6) were retrieved from The National Center of Biotechnology Information (NCBI) [22]. Interproscan v5.0 [23], with default parameters, was used for protein domain annotation and to predict domain composition. Sequences containing at least one Trx or Grx domain were extracted, and BLASTP [24] was used to confirm them as thioredoxins or glutaredoxins, as well as to classify them into the categories defined by [5]. Following [5], the Trx homologs protein disulfide isomerases (PDI) and nucleoredoxins were also considered for posterior analysis.

Based on the protein sequence information, seqinr [25] was used to compute isoelectric point and molecular weight (in kDa), while the tools tPlant-mPLoc v2.0 [26] were used to predict the subcellular localization of each protein. The GFF3 annotation file was used to determine chromosomal location and gene structure of *GmTrx* and *GmGrx* homologs. In addition, the 2000 bp genomic sequences located on the 5’ upstream of the Transcriptional Start Site (TSS) of the *GmTrx and GmGrx* sequences were extracted and analyzed with PlantCARE [27], in order to predict cis-regulatory elements (CRE) of *GmTrx* and *GmGrx.* Protein motifs were analyzed using the MEME suite [28]. The maximum motif number was set to 10, with an optimum width range set to 6-50 amino acids. The remaining parameters were kept as default. The identified motifs were annotated using Interproscan v5.0 [23]. Chromosomal distribution, *GmTrx* and *GmGrx* homolog gene structure, and protein motif, added to CRE elements, were plotted using R base functions [29] and pheatmap v1.0.12 [30]. 

For comparison purposes, Arabidopsis Trx and Grx protein sequences were retrieved from The Arabidopsis Information Resource (TAIR) [31]. For each protein family (i.e., Trx and Grx), *Glycine max* and Arabidopsis amino acid sequences were aligned using ClustalW, as implemented in the R package msa [32]. The obtained multiple sequence alignments were used as input in phangorn [33] to construct the neighbor-joining phylogenetic trees.

### 2.2. Plant Growth and Drought Assay

The experiments were carried out with Don Mario 6.8i (DM) soybean genotype. Plants were grown in a 0.5 L plastic bottle (pot) filled with a mix of sand:vermiculite (1:1) under controlled conditions in a growth chamber with a day/night cycle temperature of 28/20 °C, respectively, and a light/darkness photoperiod of 16/8 h, respectively. Relative humidity was 39.5 ± 7.7% during the entire growth period. Three seeds per pot were sown and only the healthiest seedling remained after cotyledon expansion. In addition, the seedlings’ homogeneity was carefully analyzed to avoid any interference related to developmental phenotype. For the inoculated plants, the *Bradyrhizobium elkanii* strain U1302 was used. The strain was grown in liquid YEM-medium [34].

The experimental design of the drought assay was completely randomized and consisted of four combined treatments with five biological replicates (n = 5) each, comprising a total of 20 pots. The experimental unit was 1 pot with 1 plant. The four combined treatments were nodulated (N) water-restricted (WR) plants (N+WR), nodulated well-watered (WW) plants (N+WW), non-nodulated (NN) water-restricted plants (NN+WR), and non-nodulated well-watered plants (NN+WW). During the first 19 days after sowing (V2-3 developmental stage), soybean seedlings were grown without water restriction and the substrate was kept at field capacity with B & D medium [35] supplemented with KNO_3_ (0.5 mM and 5 mM final concentration for nodulated and non-nodulated plants, respectively). From day 20 (day 0 of the water deficit period), watering was withdrawn to the WR plants while the WW plants were maintained without water restriction throughout the assay. The substrate water content was measured daily by gravimetry (water gravimetric content) during the growth and water deficit period (Appendix A). Stomatal conductance of all plants was measured on day 20 (day 0 of the water deficit period) on the abaxial leaf surface with a Porometer Model SC-1 (Decagon Device, Pullman, WA, USA), according to manufacturer instructions, to determine this parameter value at the beginning of the water deficit period. The stomatal conductance measurement was performed daily for all plants until the end of the water deficit period, which was determined individually for each water-restricted plant when the stomatal conductance value was approximately 50% of the one obtained on day 0 of the water deficit period. At the end of the water deficit period of each WR plant, the roots were harvested and kept at −80 °C until polysomal fraction purification. The roots of the WW plants were harvested together with the WR plants and also kept at −80 °C until polysomal fraction purification.

### 2.3. Polysomal Fraction Purification by Sucrose Cushion Centrifugation

#### 2.3.1. Preparation of Cytoplasmic Lysates

All steps were performed at 4 °C or in ice and all equipment and materials were pre-chilled and RNase-free. An amount of 2 mL of packed volume of frozen pulverized roots was homogenized in 4 mL of Polysome Extraction Buffer (PEB; 200 mM Tris-HCl pH 9.0, 200 mM KCl, 25 mM ethylene glycol tetraacetic acid (EGTA), 35 mM MgCl_2_, 1% detergent mix (Brij-35 20% (*w/v*), Triton X-100 20% (*v/v*), Igepal CA-630 20% (*v/v*), Tween-20 20% (*v/v*)), 1% polyoxyethylene 10 tridecyl ether (PTE), 1 mM dithiothreitol (DTT), 1 mM phenylmethylsulfonyl fluoride (PMSF), 50 μg mL^−1^ cycloheximide, 50 μg mL^−1^ chloramphenicol) using mortar and pestle. Homogenate was maintained for 15 min (or until all samples were processed) at 4 °C with gentle shaking and then clarified by centrifugation at 16,000× *g* for 15 min. Then, the homogenate was filtered with cheesecloth and the centrifugation step was repeated. An amount of 500 µL of the supernatant was reserved for isolation of total RNA (TOTAL). The remaining supernatant, 2 mL, was subjected to centrifugation through sucrose cushions for the polysomal fraction purification.

#### 2.3.2. Sucrose Cushion Centrifugation and Polysome Purification

The 2 mL clarified cytosolic extract was loaded on two layers of sucrose cushions (4.5 mL of 12% and 4.5 mL of 33.5%) and centrifuged in a Beckman L-100K class S ultracentrifuge (W40 Ti swinging bucket rotor) at 4 °C for 2 h at 35,000 rpm: 13.2 mL Ultra-Clear tubes were used (Beckman Coulter, Poway, CA, USA, 344059). The 12% and 33.5% sucrose layers were made from a 2M sucrose stock solution, a 10× salts stock solution (400 mM Tris-HCl pH 8.4, 200 mM KCl, 100 mM MgCl_2_) used at 1×, 50 μg mL^−1^ cycloheximide, and 50 μg mL^−1^ chloramphenicol. After centrifugation, the polysomal fraction was recovered as a pellet and resuspended in 200 µL of Polysome Resuspension Buffer (PRB; 200 mM Tris-HCl pH 9.0, 200 mM KCl, 25 mM EGTA, 35 mM MgCl_2_, 5 mM DTT, 50 μg mL^−1^ cycloheximide, 50 μg mL^−1^ chloramphenicol) pipetting up and down several times. The resuspended polysomal pellet was maintained for 30 min at 4 °C and then regular RNA purification was performed to obtain the polysome-associated mRNA (PAR) fraction.

### 2.4. TOTAL and PAR RNA Fraction Extraction and Transcriptome Sequencing

RNA extraction was performed with TRizol LS reagent (Invitrogen, Waltham, MA, USA, 10296-028). The resuspended polysomal pellet and the extract reserved for isolation of total RNA (2.3.1) were homogenized in 750 µL of TRizol. Samples were incubated for 5 min at room temperature (RT), followed by the addition of 200 µL of chloroform. Tubes were vigorously shaken for 15 s, incubated at RT for 10 min, and centrifuged at 4 °C, 12,000× *g* for 15 min for phase separation. Then, 500 µL from the upper phase was transferred to a new tube and 375 µL of cold isopropanol and 0.5 µL of RNase-free glycogen (Invitrogen, Waltham, MA, USA, 10814-010) were added. The mix was incubated at 4 °C for 10 min. The RNA precipitate was collected by centrifugation at 12,000× *g* for 15 min and washed with 1 mL of cold 75% ethanol. After centrifugation, the RNA pellet was air-dried, resuspended in 50 µL of Rnase-free water, and incubated at 65 °C for 5 min. RNA concentration and integrity were measured using an Agilent 2100 bioanalyzer (Agilent Technologies, Inc., Santa Clara, CA, USA). Samples with a RIN (RNA integrity number) >7.0 and >1.0 µg were sent to Macrogen Inc. (Korea) for library preparation and sequencing. TruSeq Stranded mRNA paired-end (PE) cDNA libraries were made and sequenced by the Illumina high-throughput sequencing platform. TOTAL and PAR samples from three biological replicates per combined treatment were sent for analysis.

### 2.5. Processing of Sequencing Data

Illumina sequencing data quality was visually inspected using FastQC v0.11.9 (https://www.bioinformatics.babraham.ac.uk/projects/fastqc/) (1 June 2022). Adaptors and low-quality bases were trimmed using trimmomatic [36], keeping PE reads with overall phred quality > 30 and length > 80 bp for posterior analysis.

Salmon v0.12.0, in quasi-mapping mode [37], was used for reads mapping to the *Glycine max* v4.0 transcriptome (GCF_000004515.6, retrieved from NCBI [22], and transcript abundance quantification. The estimated transcript-level abundances were converted to gene-level expression abundances using the R/BioConductor package tximport [38]. Descriptive statistics were estimated using R base functions, while differential expression analyses were performed using DESeq2 [39]. Genes with |log2FC| > 1 and adjusted *p*-value (padj)  <  0.05 were considered differentially expressed in our study. The generated gene lists were filtered to keep only differentially expressed Trx and Grx. Plots were generated using R base functions, and the R packages ggplot2 [40]. Gene list analysis was performed using STRING [41].

All sequencing data generated in this study were deposited in the NCBI Sequence Read Archive (BioSample accessions: SAMN30227622-SAMN30227645, BioProject ID: PRJNA868178).

## 3. Results

### 3.1. Trx and Grx Family Member Identification and Initial Characterization in Glycine max

A total of 187 and 101 Trx and Grx homologs, respectively, were identified from the *Glycine max* v4.0 proteome. After discarding multiple proteins codified by the same gene, a total of 125 Trx and 89 Grx homologs were considered for posterior analysis (Appendix A, respectively). For posterior analysis, the Trx proteins were consecutively named GmTrx01 to GmTrx125. Similarly, Grx were consecutively named GmGrx01 to GmGrx89. In each case, the assigned numbering follows the same order as the NCBI IDs of these proteins. Blast alignment allowed the classification of 123 out of 125 Trx and 82 out of 89 Grx in previously defined classes. Phylogenetic analysis showed a robust correspondence between GmTrx and GmGrx aggrupation with the same families of proteins in the model species Arabidopsis (Appendix A).

GmTrx homolog lengths ranged from 117 (GmTrx53 and GmTrx75) to 1067 amino acids (GmTrx114), while GmGrx showed a lower dispersion in protein length, ranging from 95 (GmGrx81) to 745 amino acids (GmGrx39). In accordance, their molecular weight ranged from 13 kDa to 116.6 kDa in GmTrx homologs and from 10.4 kDa to 83.8 kDa in GmGrx. The predicted values for the isoelectric point (pI) varied from 4.6 (GmTrx04 and GmTrx18) to 9.6 in GmTrx homologs (GmTrx103) and from 4.7 (GmGrx64) to 9.6 (GmGrx45) in GmGrx. This way, according to the predicted pI, 64 GmTrx and 32 GmGrx were acidic (pI < 7), while 61 GmTrx and 57 GmGrx were basic (pI > 7). Regarding subcellular localization, Trx homologs have a heterogeneous predicted localization, for which 49 are expected to be located in the chloroplast, 30 in cytoplasm, and 16 in the nucleus, with the remaining 30 expected to be located in the Golgi apparatus, endoplasmic reticulum, mitochondrion, cell membrane or have mixed subcellular localization. On the other hand, most GmGrx (75 out of 89) have predicted chloroplast localization.

By mapping the 125 and 89 *GmTrx* and *GmGrx* homologs on the *Glycine max* chromosomes, it can be observed that even their physical locations are unevenly distributed (Figure 1), this distribution not being correlated with chromosome length (*p* > 0.05, Spearman). There is at least one member of each family on each chromosome. A total of 22 out of 125 *GmTrx* and 24 out of 89 *GmGrx* were distributed in chromosomes arranged in clusters (i.e., with a physical distance between genes lower than 200 Kb), containing two to five genes/cluster.

### 3.2. Phylogenetic, Gene Structure, and Conserved Motif Analyses of GmTrx and GmGrx Gene Family

Phylogenetic analysis revealed that *GmTrx* genes could be divided into five subgroups (Figure 2a). The largest subgroup G2 consisted of 43 *Gm**Trx* members, whereas subgroups G1, G4, and G5 contained 27, 21, and 22 *GmTrx* members, respectively. Subgroup G3 contained 12 members.

The ten most conserved motifs for GmTrx were explored using the MEME suite and annotated using InterProScan. Four motifs (1, 2, 4, and 7) were annotated as Trx domains, which were present in most of the GmTrx (88%, 96.8%, 29.6%, and 11.2%, respectively) (Figure 2b). The analysis showed that the 89 GmTrx have a heterogeneous composition of conserved motifs, ranging from the presence of one motif in some proteins of subgroup G5 to seven motifs in the G3 subgroup.

We examined the exon–intron structures to further understand *GmTrx* genes. The results demonstrated structural variation among these *GmTrx* genes, ranging from 1 to 28 exons, whereas 26% (33/125) *GmTrx* contained 3 exons (Figure 2c). Among the *GmTrx* genes, the genes in subgroup G1 have 3 to 11 exons (Figure 2c). Subgroup G2 has one to ten exons, whereas in subgroups G3 the majority of the genes have seven exons (Figure 2c). Subgroups G4 and G5 exhibited genes with 10 to 28 exons (Figure 2c).

Phylogenetic analysis revealed that GmGrx could be divided into seven subgroups (Figure 3a). Large subgroups G3 and G6 consisted of 28 and 25 GmGrx members, respectively, whereas small subgroups G1, G2, G4, G5, and G7 contained 2, 8, 13, 8, and 5 GmGrx members, respectively.

The ten most conserved motifs for GmGrx were explored using the MEME suite and annotated using InterProScan. Five motifs (1, 2, 3, 6 and 7) were annotated as Grx domains, which were present in most of the GmGrx (42%, 71%, 14%, 57% and 14%) (Figure 3b). The analysis showed that the 89 GmGrx have a heterogeneous composition of conserved motifs, ranging from the presence of one motif in subgroup G1 to four motifs in subgroups G6 and G7.

To gain more insight into *GmGrx* genes, we examined the exon–intron structures. The results demonstrated structural variation among these *GmGrx* genes, ranging from one to seven exons, whereas 62% (55/89) of *GmGrx* contained one exon (Figure 3c). Among the *GmGrx* genes, all the members of subgroup G1 have three exons (Figure 3c). Subgroup G2 has two to six exons, whereas subgroups G3 and G4 contain fewer exons (between one and two) (Figure 3c). Subgroups G5 and G7 are richer in exons (from one to seven). Generally, *GmGrx* genes in the same subgroup exhibited similar exon–intron features, providing further evidence of their phylogenetic relationships.

### 3.3. Cis-Regulatory Elements (CRE) for GmTrx and GmGrx

To understand the potential regulatory mechanisms of *GmTrx* and *GmGrx* genes, we analyzed the presence of cis-regulatory elements 2000 bp upstream the TSS. Based on their putative functions, the identified cis-acting elements were further classified into five distinct groups (Appendix A). Except for the common cis-acting elements (such as enhancer element CAAT-box and core promoter element TATA-box), the most abundant elements were stress-responsive elements, including WUN motifs that were present in 56% and 61% of *GmTrx* and *GmGrx*, respectively, TC-rich repeats present in around 50% of *GmTrx* and *GmGrx*, an MYB binding site involved in drought inducibility (MBS) present in 40% of the *GmTrx* and *GmGrx*, and the anaerobic induction ARE motif present in 72% of the *GmTrx* and 67% of the *GmGrx* putative promoters. Some CRE were involved in light responsiveness, such as the G-box present in 73% and 78% of the *GmTrx* and *GmGrx* promoter regions, respectively. We found a high frequency of light-responsive elements both in *GmTrx* and *GmGrx*, with Box IV being present in 95% of the *GmTrx* and 94% of the *GmGrx* and G-box present in 75% in both gene classes. Moreover, we detected hormone-related cis-acting elements, including the MeJA-responsive element (TGACG motif and CGTCA motif), the SA-responsive element (TCA element), and the ABA-responsive element (ABRE) as the more widespread among the *GmTrx* and *GmGrx*. This suggests that these genes are regulated by light, abiotic stresses, and hormones.

### 3.4. Identification of the Differentially Expressed GmTrx and GmGrx Genes in Nodulated and Water-Restricted Soybean Plants

In our experimental design, Don Mario soybean plants were subject to four combined treatments comprising nodulation (nodulated and non-nodulated plants) and water-deficit conditions (water-restricted and well-watered plants). The contrasts between the combined treatments analyzed in this study were N+WR vs. N+WW and N+WR vs. NN+WR. TOTAL and PAR RNA fractions were obtained and analyzed to study the transcriptome and translatome of the treated plants.

Both contrasts evidence the differential response of nodulated and water-restricted plants with respect to the different control conditions, i.e., well-watered plants in the first and non-nodulated plants in the second. The first contrast (N+WR vs. N+WW) shows the response of nodulated plants to water deficit, while the second contrast (N+WR vs. NN+WR) shows the distinctive response to water deficit of nodulated plants regarding non-nodulated plants. Since we analyzed both TOTAL and PAR RNA fractions, the plant responses to water deficit in the two possible nodulation conditions were further classified according to their transcriptional, translational, or mixed transcriptional plus translational regulation.

When analyzing the total number of DEGs in both contrasts, we found that it was higher in the first one at all levels, i.e., TOTAL, PAR, and TOTAL+PAR. The imposition of water deficit altered the expression of a greater number of DEGs than the condition of plant nodulation. Moreover, most DEGs were at the TOTAL+PAR level, showing that most genes are regulated at the transcriptional and translational levels. The number of DEGs, classified in up- and down-regulated ones, for both contrasts and for the TOTAL and PAR factions is shown in Figure 4a. The Venn analysis of the N+WR vs. N+WW contrast indicated that more DEGs are down-regulated than up-regulated in TOTAL (597/406), PAR (327/277), and TOTAL+PAR (1126/870; intersection TOTAL-PAR). However, the N+WR vs. NN+WR contrast analysis showed that while for TOTAL and PAR there were more down-regulated DEGs than up-regulated (144/62 and 44/36, respectively; Figure 4a), for TOTAL+PAR the majority of DEGs were up-regulated (242/123). Although the majority of DEGs were found to be regulated at the transcriptional and translational levels (TOTAL+PAR), it is interesting to notice that in both contrasts, and in up- and down-regulated DEGs, there were genes exclusively regulated at the translational level (PAR) (Figure 4a). The translational control of gene expression allows cells to respond to a stimulus quickly, providing flexibility and adaptability.

Among these large DEG lists, we focused our analysis on the *GmTrx* and *GmGrx* genes due to the relevant role of the Trx and Grx system in dealing with ROS homeostasis in organisms subjected to stressful conditions. A total of 112 out of 125 predicted *GmTrx* were observed in our RNA-seq data. Regarding the *GmGrx*, 62 out of 89 were detected. Expression data of these genes, in both contrasts and in the TOTAL and PAR RNA fractions, are depicted in Figure 4b,c. As expected, the expression profiles of the TOTAL and PAR fractions of each contrast showed a high correspondence between them. In addition, in these heatmaps, it became apparent that a significant fraction of *GmTrx* and *GmGrx* showed differential expression levels in the conditions assayed herein, with both transcriptional and translational regulation. This way, a total of 12 *GmTrx* and 12 *GmGrx* (i.e., 10.7% of the detected *GmTrx* and 19% of the detected *GmGrx*) were differentially expressed in nodulated and water-restricted plants considering both contrasts and RNA fractions (indicated with asterisks in the profile -or profiles- in which the DEG condition was achieved, in Figure 4b,c, respectively).

From the 12 *GmTrx* that were differentially expressed, 6 belong to the typical and 6 to the atypical types described for Arabidopsis. The typical ones correspond to the largest class, the h class, with three h II, two h II, and one h I members. The atypical ones, which mostly await elucidation, comprise the nucleoredoxin, TTL, and Lilium classes with four, one, and one members, respectively (Table 1). All of them were DEGs at the N+WR vs. N+WW contrast. Furthermore, *GmTrx48* and *GmTrx23* were DEG at the N+WR vs. NN+WR contrast. The regulation level, i.e., the RNA fraction (TOTAL, PAR, or TOTAL+PAR) at which they are DEGs, and the up-regulated or down-regulated status are also specified in Table 1. *GmTrx92* (TTL class) is the only one exclusively regulated (down-regulated) at the translational level (PAR RNA fraction).

In the case of the 12 DEG *GmGrx*, only the CCxS and 4CxxC classes described for Arabidopsis were present, with five and seven members of the CCxS and 4CxxC classes, respectively (Table 2). Similar to *GmTrx*, all *GmGrx* were DEGs at the N+WR vs. N+WW contrast; for *GmGrx37*, *GmGrx38*, and *GmGrx 83*, the DEG condition was also achieved in the N+WR vs. NN+WR contrast. Nine out of twelve DEG *GmGrx* were down-regulated. *GmGrx43* and *GmGrx50*, both members of the 4CxxC class, were exclusively regulated (down-regulated) at the translational level.

To gain more insight into the functionality of the DEGs, we analyzed the distribution and localization of the most frequent *GmTrx* and *GmGrx* CREs (stress response: ARE and WUN motif; site-binding-related: MYC and MYB; hormone response: CGTCA, TGACG, TCA, and ABRE; and light response: TCT and Box IV) in the promoter regions of the DEGs. Interestingly, the promoters of the *GmTrx* and *GmGrx* DEGs varied in composition, position, and number of the CREs analyzed. For example, the *GmTrx* and *GmGrx* DEGs have one to two WUN motifs in their promoters and one to four ARE motifs. The WUN motif was present in five of the *GmTrx* and in six of the *GmGrx* DEGs, while the ARE motif was present in all the promoters analyzed (Figure 5). Box IV was present in all the promoter regions analyzed and varied in number (from 1 to 12) and position, independent of the protein class type (Figure 5).

## 4. Discussion

The virtually complete reference genome of *Glycine max,* one of the most important oilseed crops in the world, was published in 2009. Since then, several contributions have been made by the soybean community to improve genome annotation and characterization. However, there are still a substantial number of proteins with no assigned functions: from the total proteins annotated from the reference genome, ~25% belong to the category “uncharacterized proteins”. This is even more exacerbated in some family proteins, for which the percentage of proteins in this uncertain category is even higher. In an effort to contribute to the characterization of the annotated genome, here we performed the initial identification and description of the complete repertoire of thioredoxins (Trx) and glutaredoxins (Grx) in *Glycine max*.

The classification and naming of the *Trx* and *Grx* gene family is complex [5] and can be progressively improved with the identification of these families in more species [9,42,43,44]. Overall, phylogenetic analysis showed a robust correspondence between GmTrx and GmGrx aggrupation with the same families of proteins in Arabidopsis. Based on the type of domain related to protein function, we identified 125 Trx and 89 Grx genes in soybean (Appendix A, respectively). *GmTrx* genes were divided into five subgroups, whereas *GmGrx* genes were divided into seven subgroups by phylogenetic analysis (Figure 2a). We found that the distribution of some GmTrx and GmGrx was disordered; some GmTrx classified in different classes such as O and h were found in the same subgroup. Moreover, some GmTrx classified in the Lilium class were found in two distinct subgroups; the same was observed with some GmGrx with the CCxS motif, which were found in two different GmGrx subgroups (Figure 2 and Figure 3; Appendix A). The previously mentioned results were also reported in Vitis [44] and could be due to the diversity of domain types besides the Trx and Grx domains.

The distribution and type of CREs in promoters determine gene activities and functions. In this study, through a systematic analysis of CRE in the promoter regions of *GmTrx* and *GmGrx*, we identified various types of CRE (Appendix A). Related to abiotic stress, we found that a high number of promoters of both *GmTrx* and *GmGrx* have the WUN motif, TC-rich repeats, the MYB binding site involved in drought inducibility (MBS) [45], and the anaerobic induction ARE motif [46]. In addition, most promoters have the light-responsive elements G-box [47] and Box IV [48]. Moreover, we detected hormone-related cis-acting elements, and the more widespread were the MeJA-responsive element (TGACG motif and CGTCA motif) [49], the SA-responsive element (TCA element) [50,51], and the ABA-responsive element (ABRE) [52,53]. This suggests that these genes are regulated by light, abiotic stresses, and hormones.

The translational control of gene expression is widely used in different biological situations [54,55]. Plants and other eukaryotic organisms benefit from this control step in cases that require a rapid response to a stimulus, since it does not require the de novo synthesis of mRNAs but rather relies on the efficiency with which the mRNAs already present in the cells are translated [56,57,58,59]. This characteristic of translational control provides flexibility and adaptability to the organisms when facing environmental changes such as water restriction. Under this situation, and others that imply a reduction in energy availability (e.g., hypoxia) or nutrient shortage, a general repression of translation occurs, affecting the majority of cellular mRNAs. However, specific mRNAs increase their association with polysomes under the afore-mentioned conditions [60,61,62,63]. In plants under water-deficit stress, dehydration-inducible genes such as those coding for dehydrins and late embryogenesis abundant (LEA) proteins are examples of messengers with enhanced translation rates [60]. These proteins are necessary for survival or adaptation to suboptimal growth conditions [59]. Specifically, in legume plants, the BNF process is highly susceptible to water-deficit stress, and there is evidence suggesting that the nodulation condition of the plant affects its response strategies to water-deficit stress [64,65,66,67]. Moreover, it has been shown that, although translation is not globally affected after rhizobium infection, specific mRNAs that code for proteins of the Nod signaling pathway (i.e., Nod factor receptors and transcription factors) are selectively recruited to polysomes [68]. Our experimental assay was designed to analyze the transcriptional and translational response of nodulated and water-restricted plants by contrasting this combined treatment condition (N+WR) with two other combined treatment conditions, comprising nodulated and well-watered plants (N+WW) and non-nodulated and water-restricted plants (NN+WR). Overall, we found that the condition that most altered gene expression was water deficit and that, as expected, most DEGs obtained in both contrasts were regulated at the transcriptional and translational (TOTAL+PAR) levels [19,67] (Figure 4a). Most DEGs related to the response of nodulated plants to water deficit (obtained in the N+WR vs. N+WW contrast) were down-regulated at all levels, suggesting a general repression of gene expression (Figure 4a). However, the specific responses to water deficit of nodulated plants regarding non-nodulated plants (N+WR vs. NN+WR contrast) comprised DEGs that were mostly up-regulated at TOTAL+PAR levels (Figure 4a). These genes include nodulins, proteins with both metabolic and structural roles induced during the BNF process [69]. Interestingly, and in accordance with previous studies [21,63,68,70,71,72], in both contrasts, we found DEGs that belong exclusively to the PAR fraction, i.e., genes that changed (increased for the up-regulated or decreased for the down-regulated) their association to polysomes due to water-deficit stress or the different nodulation condition (Figure 4a). Although transcriptional reprogramming is crucial to the response to environmental perturbations in eukaryotes, translational regulation contributes to adaptation and survival by limiting consumption of ATP and directing the synthesis of specific proteins [71].

In this context, both Trx and Grx proteins have been proven to have key roles in fine-tuning the ROS levels in different stress responses and participating in dynamic processes that the plant requires to acclimate and adapt to the changing environment [73,74,75]. Our analysis regarding the poorly characterized soybean Trx and Grx showed that a significant fraction (approximately 10% and 20%, respectively) of the detected *GmTrx* and *GmGrx* in our RNA-seq data were DEGs in nodulated and water-restricted plants, suggesting a relevant role of these proteins in the plant response to water-deficit stress in nodulation conditions (Figure 4b,c; Table 1; Table 2). Moreover, from the 12 DEG *GmTrx*, *GmTrx92* was exclusively regulated, specifically down-regulated, at the translational level (Figure 4b; Table 1). This protein had high homology with Arabidopsis TTL1 (Appendix A) for which a role in the root adaptation during osmotic stress is suggested [76]. It is worth noting that four out of five soybean nucleoredoxins were DEGs in nodulated and water-restricted plants; specifically, they were up-regulated at the TOTAL+PAR level in the N+WR vs. N+WW contrast (Figure 4b; Table 1). These proteins presented high homology with the Arabidopsis nucleoredoxins (NRX1 and NXR2; Appendix A). In particular, NRX1 was shown to target antioxidant enzymes, such as catalases, which suffer oxidative distress in ROS-rich environments and require reductive protection for optimal activity [77]. Moreover, *GmTrx74*, *GmTrx52*, and *GmTx64* were also found to be overexpressed in a transcriptomic assay performed in leaves of soybean plants subjected to drought [78]. In the case of *Grx*, from the 12 DEG *GmGrx, GmGrx43* and *GmGrx50* were exclusively regulated, and also down-regulated, at the translational level. To our knowledge, this is the first time that translational regulation (derived from a polysome profiling approach) is reported for any member of the Trx/Grx system. GmGrx01, 15, 20, 38, 47, 79, and 83, all CCxS class, presented high homology with Arabidopsis ROXY proteins 4, 5, 19, and 21 (Appendix A). These proteins are CC-type Grx that interact with the TGA2 (TGACG-binding) factor [79] and suppress the ethylene-responsive factor 59 (ORA59) promoter activity [80,81]. Five out of seven GmGrx from the CCxS class have the TGACG binding domain.

As observed for all DEGs, most DEG *GmTrx* and *GmGrx* were regulated at both transcriptional and translational levels. Furthermore, as mentioned before, *GmTrx92*, *GmGrx43*, and *GmGrx50* were exclusively regulated at the PAR level, suggesting that the translational control of these genes is relevant.

By constructing protein–protein interaction networks using STRING, we further analyzed possible targets of our DEG GmTrx and GmGrx among the DEG lists obtained in both contrasts and at each regulation level (TOTAL, PAR, TOTAL+PAR) (Figure 4a). In the down-regulated differentially expressed genes at the TOTAL+PAR levels in the N+WR vs. NN+WR contrast, we found an enriched network, i.e., with significantly more interactions than expected, in which GmTrx23 was, on one hand, associated through several types of interactions (known, predicted, and others such as text-mining and co-expression) to two nitrate reductase (NR) enzymes (Figure 6). This suggests that INR2 and NP_001345469 (an NAD(P)H-dependent isoform) NRs could be targets of the class h III GmTrx23. Moreover, these NR enzymes were associated with other proteins related to nitrogen (N) metabolism (nitrate high-affinity transporters and a ferredoxin-dependent NR), explaining the network enrichment in the functional term “nitrogen metabolism (gmx00910)” of KEGG pathway (Figure 6, proteins colored in red). On the other hand, GmTrx23 directly interacted with GmGrx38. This could be a case of the Trx alternative reduction pathway, in which Trx are targets of Grx activity [5]. The other over-represented functional term in the network was the biological (Gene Ontology, GO) “oxidation-reduction process (GO:0055114)”. Both GmTrx and GmGrx, along with eight other proteins, were included in this term (Figure 6; proteins colored in blue). Somehow, the down-regulation of proteins related to nitrate uptake and reduction in nodulated and water-restricted plants (regarding non-nodulated and nitrate-fed plants) can be expected. However, it was not so obvious to expect the same for GmTrx23 and GmGrx38, proteins that are also down-regulated in the N+WR vs. N+WW contrast, implying that this is exclusively a water-deficit stress response, taking into account the known roles of Trx and Grx proteins as key players for redox regulation during water-deficit stress. Furthermore, it can be hypothesized that the Trx and Grx downregulation is a consequence of the downregulation of the proteins related to N metabolism.

## 5. Conclusions

In this study, we identified 125 *Trx* and 89 *Grx* genes in the *Glycine max* v4.0 proteome. From the total detected *GmTrx* and *GmGrx* genes in our RNA-seq experiment, 10% and 20%, respectively, were differentially expressed in nodulated and water-restricted plants, suggesting a relevant role of these proteins in the plant response to water-deficit stress in nodulation conditions. Among them, only *GmTrx92*, *GmGrx43*, and *GmGrx50* were exclusively regulated at the PAR level, suggesting that the translational control of these genes is relevant. We were able to find an enriched network in the down-regulated differentially expressed genes at the TOTAL+PAR levels in the N+WR vs. NN+WR contrast, in which GmTrx23 was, on one hand, associated through several types of interactions to two NR enzymes, and on the other hand GmTrx23 directly interacted with GmGrx38. This could be a case of Grx-dependent Trx reduction that must be resolved by future wet laboratory experiments.

## Figures and Tables

**Figure 1 antioxidants-11-01622-f001:**
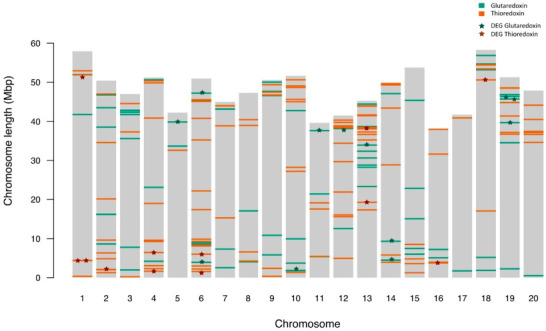
Distribution of the 125 *GmTrx* and 89 *GmGrx* gene family members on *Glycine max* chromosomes. The color of each gene indicates the corresponding family. Asterisks indicate the differentially expressed *GmTrx and GmGrx* genes.

**Figure 2 antioxidants-11-01622-f002:**
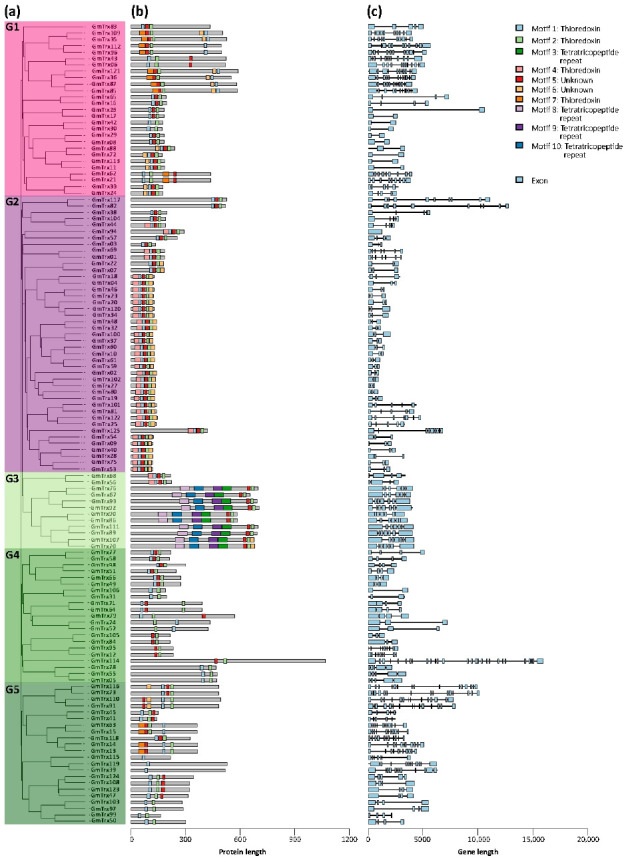
Phylogenetic tree, conserved motif, and gene structure of *Glycine max Trx* genes (*GmTrx*). (**a**) Phylogenetic relationship among the GmTrx based on the amino acid sequence alignment. (**b**) Conserved motifs in amino acid sequence of different subgroups of GmTrx. The different colored boxes on the right represent diverse conserved motifs. The general characteristics of the motifs can be found in Appendix A. (**c**) Exon–intron analysis of *GmTrx* genes.

**Figure 3 antioxidants-11-01622-f003:**
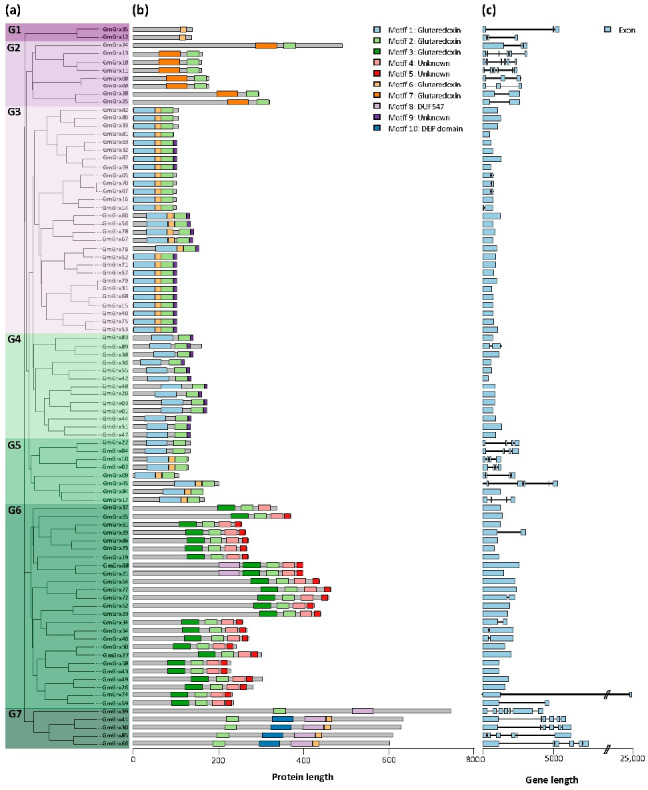
Phylogenetic tree, conserved motif, and gene structure of *Glycine max Grx* genes (*GmGrx*). (**a**) Phylogenetic relationship among the GmGrx based on the amino acid sequence alignment. (**b**) Conserved motifs in amino acid sequence of different subgroups of GmGrx. The different colored boxes on the right represent diverse conserved motifs as specified. The general characteristics of the motifs can be found in Appendix A. (**c**) Exon–intron analysis of *GmGrx* genes.

**Figure 4 antioxidants-11-01622-f004:**
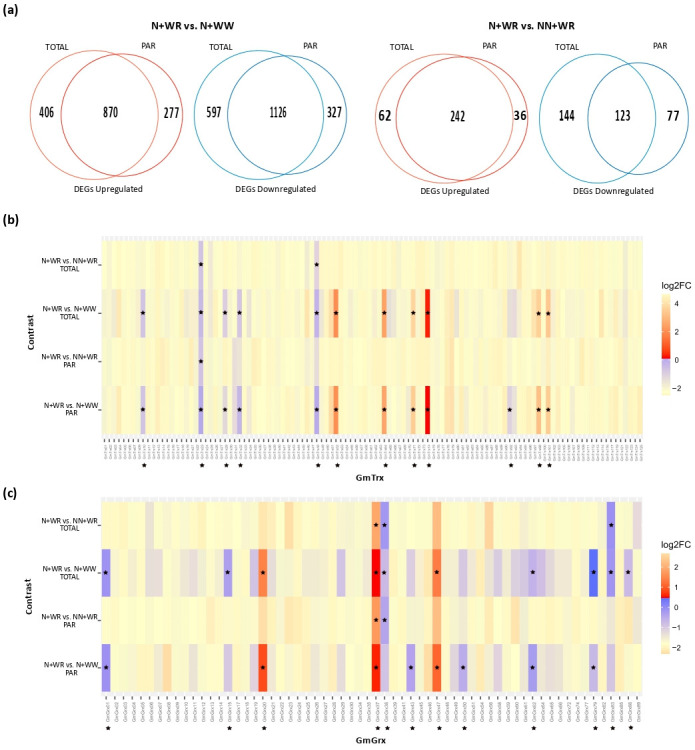
Differentially expressed gene (DEG) analysis and *GmTrx* and *GmGrx* expression profiles in nodulated (N) and water-restricted (WR) plants with respect to well-watered (WW) and non-nodulated (NN) plants. (**a**) Venn diagrams showing up- and down-regulated genes in the N+WR vs. N+WW and N+WR vs. NN+WR contrasts in total RNA (TOTAL) and polysome-associated mRNA (PAR) fractions. (**b**) Expression profiles of *GmTrx*. (**c**) Expression profiles of *GmGrx*. Heatmaps were constructed from the RNA-seq experimental data. Asterisks indicate the differentially expressed *GmTrx* and *GmGrx* genes found in our study. Genes with |log2FC| > 1 and adjusted *p*-value (padj)  <  0.05 were considered differentially expressed.

**Figure 5 antioxidants-11-01622-f005:**
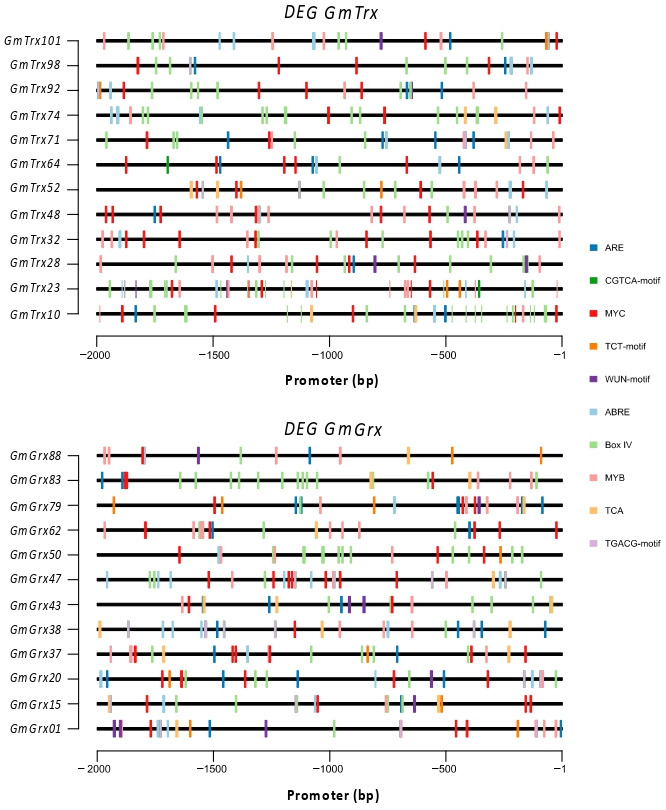
Number and localization of the more frequent cis-acting elements among the *GmTrx* and *GmGrx* family in the promoter regions of the DEGs. The 2000 bp genomic upstream of the Transcriptional Start Site (TSS) of *GmTrx* and *GmGrx* sequences were analyzed with PlantCARE [27], in order to predict their cis-regulatory elements (CRE). The cis-acting elements presented are labeled with different colors and illustrated on the right side. The TATA-box and CAAT-box are not shown.

**Figure 6 antioxidants-11-01622-f006:**
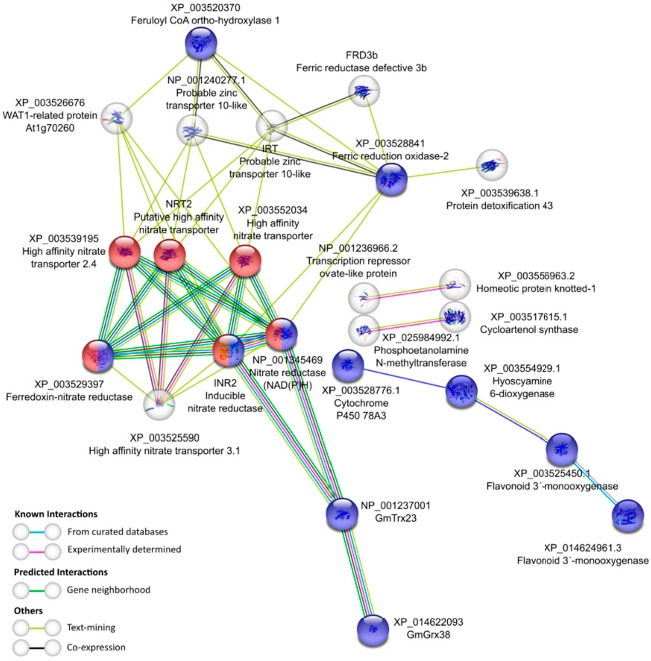
String analysis [41] derived protein-protein interaction network obtained for the differentially expressed genes down-regulated at the TOTAL+PAR level in nodulated and water-restricted plants with respect to non-nodulated and water-restricted plants (N+WR vs. NN+WR contrast). The network nodes represent proteins. Connecting lines denote protein-protein associations whose colors represent the type of interaction evidence as specified in the legend. Red proteins: KEGG Pathway functional term “Nitrogen metabolism (Gmx00901)”; Blue proteins: GO functional term “Oxidation-reduction process (GO:0055114)”. Disconnected nodes, i.e., proteins not showing any interaction in the network, were deleted.

**Table 1 antioxidants-11-01622-t001:** Differentially expressed *GmTrx* in nodulated and water-restricted plants. The class, chromosome localization, subcellular putative localization, and protein length are shown. The contrast (N+WR vs. N+WW and/or N+WR vs. NN+WR) in which the DEG condition was achieved together with the status (up- or down-regulated) and the regulation level (TOTAL, PAR, or TOTAL+PAR) are also shown.

ProtID_Trx	GmTrx	Class	Chromosome #	Putative Localization	Length (aa)	Condition DEG	Status	Regulation Level
NP_001237762.1	GmTrx32	h II	1	Cytoplasm.	138	N+WR vs. N+WW	Downregulated	TOTAL+PAR
NP_001236052.1	GmTrx10	h II	1	Cytoplasm.	126	N+WR vs. N+WW	Downregulated	TOTAL+PAR
NP_001237535.2	GmTrx28	h I	1	Cytoplasm.	120	N+WR vs. N+WW	Downregulated	TOTAL+PAR
NP_001240862.1	GmTrx48	h II	2	Cytoplasm.	138	N+WR vs. N+WW	Downregulated	TOTAL+PAR
						N+WR vs. NN+WR		TOTAL
XP_003522672.1	GmTrx64	Nucleoredoxin	4	Chloroplast. Cytoplasm. Nucleus.	389	N+WR vs. N+WW	Upregulated	TOTAL+PAR
NP_001276271.2	GmTrx52	Nucleoredoxin	4	Cytoplasm.	423	N+WR vs. N+WW	Upregulated	TOTAL+PAR
XP_003527521.1	GmTrx74	Nucleoredoxin	6	Cytoplasm. Nucleus.	434	N+WR vs. N+WW	Upregulated	TOTAL+PAR
XP_003526462.1	GmTrx71	Nucleoredoxin	6	Nucleus.	389	N+WR vs. N+WW	Upregulated	TOTAL+PAR
NP_001237001.1	GmTrx23	h III	13	Chloroplast. Cytoplasm.	122	N+WR vs. N+WW	Downregulated	TOTAL+PAR
						N+WR vs. NN+WR		
XP_003543765.1	GmTrx92	TTL	13	Nucleus.	703	N+WR vs. N+WW	Downregulated	PAR
XP_003548763.1	GmTrx98	Lilium	16	Chloroplast.	299	N+WR vs. N+WW	Upregulated	TOTAL+PAR
XP_003552324.1	GmTrx101	h III	18	Cytoplasm.	139	N+WR vs. N+WW	Upregulated	TOTAL+PAR

**Table 2 antioxidants-11-01622-t002:** Differentially expressed *GmGrx* in nodulated and water-restricted plants. The class, chromosome localization, subcellular putative localization, and protein length are shown. The contrast (N+WR vs. N+WW and/or N+WR vs. NN+WR) in which the DEG condition was achieved together with the status (up- or down-regulated) and the regulation level (TOTAL, PAR, or TOTAL+PAR) are also shown.

ProtID_Grx	GmGrx	Class	Chromosome #	Putative Localization	Length (aa)	Condition DEG	Status	Regulation Level
XP_003525338.1	GmGrx37	4CxxC	5.	Chloroplast.	337.	N+WR vs. N+WW	Upregulated	TOTAL+PAR
						N+WR vs. NN+WR		
XP_003526012.1	GmGrx38	CCxS	6	Chloroplast.	140	N+WR vs. N+WW	Downregulated	TOTAL
						N+WR vs. NN+WR		TOTAL+PAR
NP_001235171.1	GmGrx01	CCxS	6	Chloroplast.	172	N+WR vs. N+WW	Downregulated	TOTAL+PAR
NP_001238068.1	GmGrx15	CCxS	10	Chloroplast.	102	N+WR vs. N+WW	Downregulated	TOTAL
XP_003537494.1	GmGrx43	4CxxC	11	Chloroplast. Nucleus.	229	N+WR vs. N+WW	Downregulated	PAR
NP_001240908.1	GmGrx20	CCxS	12	Chloroplast.	160	N+WR vs. N+WW	Upregulated	TOTAL+PAR
XP_003543019.1	GmGrx47	CCxS	13	Chloroplast.	133	N+WR vs. N+WW	Upregulated	TOTAL+PAR
XP_003545225.1	GmGrx50	4CxxC	14	Chloroplast.	242	N+WR vs. N+WW	Downregulated	PAR
XP_014622093.1	GmGrx83	CCxS	14	Chloroplast.	141	N+WR vs. N+WW	Downregulated	TOTAL
						N+WR vs. NN+WR		
XP_003554133.1	GmGrx62	4CxxC	19	Chloroplast. Nucleus.	424	N+WR vs. N+WW	Downregulated	TOTAL+PAR
XP_040868600.1	GmGrx88	4CxxC	19	Chloroplast. Nucleus.	398	N+WR vs. N+WW	Downregulated	TOTAL
XP_006604701.1	GmGrx79	CCxS	19	Chloroplast.	102	N+WR vs. N+WW	Downregulated	TOTAL+PAR

## Data Availability

All sequencing data generated in this study were deposited in the NCBI Sequence Read Archive (BioSample accessions: SAMN30227622-SAMN30227645, BioProject ID: PRJNA868178).

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
