# Peer review of "Analysis of Thioredoxins and Glutaredoxins in Soybean: Evidence of Translational Regulation under Water Restriction"

_antioxidants, 2022, doi:10.3390/antiox11081622_

Round 1

Reviewer 1 Report

The manuscript entitled “Analysis of Thioredoxins and Glutaredoxins in soybean: Evidence of Translational regulation under water restriction” by Sainz et al aimed to study the differential response of nodulated soybean plants under the water deficit stress. 

The authors conducted an RNA-seq experiment with soybean roots under the hydric and nodulation conditions, and performed initial identification and description of the complete repertoire of thioredoxins (Trx) and glutaredoxins (Grx) in soybean. They found a larger number of differentially expressed genes under water deficit conditions compared to the condition of plant nodulation. Further, they identified a network involved in the interaction between GmTrx and GmGrx, which is associated with nitrogen metabolisms. These results provide insights into the identification and characterization of the repertoire of Trx and Grx in soybean. 

Overall, the method used in the study is thorough. Conclusions are appropriate, and supported by the data. The whole study is sound, and I recommend accepting it. 

Author Response

We would like to thank reviewer 1 for the appreciation of the work that has been done in the manuscript.

Reviewer 2 Report

This manuscript entitled "Analysis Of Thioredoxins And Glutaredoxins In Soybean: Evidence Of Translational Regulation Under Water Restriction" is presenting a great work concerning the identification of differentially expressed thioredoxins and glutaredoxins in soybean. The study is well done, and leads to the characterisation of specific differentially expressed Trx and Grx in soybean.

Minor modification needs:

Line 62: Why the authors are not presenting Trxs in the introduction. ref: Ribeiro C.W., Baldacci-Cresp F., Pierre O., Larousse M., Benyamina S., Lambert A., Hopkins J., Castella C., Cazareth J., Alloing G., Boncompagni E., Couturier J., Mergaert P., Gamas P., Rouhier N., Montrichard F. and Frendo P. Regulation of Differentiation of Nitrogen-Fixing Bacteria by Microsymbiont Targeting of Plant Thioredoxin s1. Curr. Biol. DOI: 10.1016/j.cub.2016.11.013.

Lines 205-217: Change ml into mL

Line 251: I do not understand the sentence, data is submitted or not !

Tables 1 and 2: Change Localization in column 5 to Putative localization.

Figure 2: for motif 8, need to correct "Tetratricopeptide" on the side legend.

Figure 6: To my opinion, I don't know if we need to keep this figure in the paper. Maybe you should move the figure to supplemental data.

Supp. Fig 6: In a, motifs 3,4, 8,9,10 are not convincing at all. In b, motifs 1, 3, 7 and 8, are not convincing using MEME suite in my opinion.

Author Response

We thank reviewer#2 for the comments. 

Minor modification needs:

Line 62: Why the authors are not presenting Trxs in the introduction. ref: Ribeiro C.W., Baldacci-Cresp F., Pierre O., Larousse M., Benyamina S., Lambert A., Hopkins J., Castella C., Cazareth J., Alloing G., Boncompagni E., Couturier J., Mergaert P., Gamas P., Rouhier N., Montrichard F. and Frendo P. Regulation of Differentiation of Nitrogen-Fixing Bacteria by Microsymbiont Targeting of Plant Thioredoxin s1. Curr. Biol. DOI: 10.1016/j.cub.2016.11.013.

Response: We added a phrase in Line 81 about the Trx s type found in M. truncatula: “In M. truncatula there is an additional Trx type, called Trxs s, that comprise four isoforms that are associated with symbiosis.”

Lines 205-217: Change ml into mL

Response: Done, we changed ml into mL in Lines 205-217.

Line 251: I do not understand the sentence, data is submitted or not !

Response: Line 253: All sequencing data generated in this study was deposited at the NCBI Sequence Read Archive (BioSample accessions: SAMN30227622-SAMN30227645, BioProject ID: PRJNA868178). Also included in Data availability statement, Line 687.

Tables 1 and 2: Change Localization in column 5 to Putative localization.

Response: Done, we changed Localization in column 5 to Putative localization in Table 1 and 2. We have uploaded new tables.

Figure 2: for motif 8, need to correct "Tetratricopeptide" on the side legend.

Response: Done, we corrected the word Tetratricopeptide on the side legend of Figure 2. We have uploaded a new Figure.

Figure 6: To my opinion, I don't know if we need to keep this figure in the paper. Maybe you should move the figure to supplemental data.

Response: We would like to keep this figure in the paper since it shows the existence of a network, with the support of experimental evidence, and suggest the control of Trx-Grx system (that have transcriptional and translational control in hydric stress) of enzymes of nitrogen metabolism (nitrate reductases) relevant to the model that is under study.

Supp. Fig 6: In a, motifs 3,4, 8,9,10 are not convincing at all. In b, motifs 1, 3, 7 and 8, are not convincing using MEME suite in my opinion.

Response: We have changed the Supp. Fig 6 for Table S3 showing the motifs, the p-values, and the number of proteins in which the motif was found.
